# Wood and Its Impact on Humans and Environment Quality in Health Care Facilities

**DOI:** 10.3390/ijerph16183496

**Published:** 2019-09-19

**Authors:** Veronika Kotradyova, Erik Vavrinsky, Barbora Kalinakova, Dominik Petro, Katarina Jansakova, Martin Boles, Helena Svobodova

**Affiliations:** 1Institute of Interior and Exhibition Design, BCDlab, Faculty of Architecture, Slovak University of Technology, Nam. Slobody 19, 81245 Bratislava, Slovakia; martin.boles@stuba.sk; 2Institute of Electronics and Photonics, Faculty of Electrical Engineering and Information Technology, Slovak University of Technology, Ilkovicova 3, 81219 Bratislava, Slovakia; erik.vavrinsky@stuba.sk; 3Institute of Medical Physics, Biophysics, Informatics and Telemedicine, Faculty of Medicine, Comenius University, Sasinkova 2, 81372 Bratislava, Slovakia; helena.svobodova@fmed.uniba.sk; 4Institute of Biochemistry and Microbiology, Faculty of Chemical and Food Technology, Slovak University of Technology, Radlinskeho 9, 81237 Bratislava, Slovakia; barbora.kalinakova@stuba.sk; 5FaceMedia, Jazdecka 6, 83103 Bratislava, Slovakia; dominik@facemedia.io; 6Institute of Physiology, Faculty of Medicine, Comenius University, Sasinkova 2, 81372 Bratislava, Slovakia; katarina.jansakova@fmed.uniba.sk; 7Department of Simulation and Virtual Medical Education, Faculty of Medicine, Comenius University, Sasinkova 4, 81272 Bratislava, Slovakia

**Keywords:** natural materials, wood, environment design, hospitals, human physiology, face emotions

## Abstract

The paper presents the application of natural materials, especially wood, which are relevant for human well-being in built environments of health, social, and day care facilities. These properties were tested by a complex methodology in a case study in the wooden waiting room at National Oncology Institute in Bratislava. In this space, experimental tests of physiological responses were further executed on 50 volunteers moving in the waiting room for 20 min. In this article, the EEG (electroencephalograph) (four persons) and emotions from the faces of all our volunteers before entering and after a stay in a wooden waiting room were recorded. Specifically, the ECG (electrocardiograph), heart rate (HR), and respiration activity were measured by using our own designed ECG holter (40 persons), and also blood pressure and cortisol levels were observed. The usage of wooden materials verifies their regenerative and positive impact on the human nervous system, through the appealing aesthetics (color, texture, and structures), high contact comfort, pleasant smell, possibility to regulate air humidity, volatile organic compound emissions (VOC-emissions), and acoustic well-being in the space.

## 1. Introduction

Due to the aging of the population, reduced natality, and decreasing mortality connected with progress in health care in most EU and other developed countries, there is a strong need to explore the characteristics of the built environments of health care, social, and day care facilities. Even though concepts of home care are widely supported, acute or long-lasting health problems will still be managed in the hospitals and clinics of different health care facilities.

Among architectural typologies, the design of hospitals is considered one of the most complex commissions for architects. In terms of honorarium levels for architects, hospitals can be found in the highest ranks, the fourth or fifth level. Renovation of any building adds 20% to the final honorarium for architects. Today, most public commissions come from public procurements. It cannot be expected that good, recognized, and award-winning architects would work on such a commission. Construction engineers belong to the Slovak Chamber of Architects and can compete in those procurements. Engineers are not educated to bring architectural quality to the projects and can work only on structural and technological issues. They have the advantage of delivering projects for lower prices. Therefore, it is difficult to achieve complex quality under these conditions.

The majority of Slovakian hospitals and other health care facilities were built in the time of socialism, and few of them have gone through complex renovations [1]. Grant systems are set to bring measurable outcomes, such as the energy efficiency of buildings. This is addressed by generous additional thermal insulation of buildings, new external plastering of facades, and replacement of technologies. Very few interiors of these buildings have been complexly and programmatically renovated. Hospitals in Slovakia are evaluated negatively in terms of their cold and impersonal environments, monotonous and depressive interiors, non-logical layouts, lack of non-barrier solutions, archaic interior facilities, inappropriate illumination, noise, etc. These spaces do not reflect qualities, such as safety, familiarity, or client-friendly treatment, but end in disorientation and stress [1]. Across the entire world, visible trends can be observed of hospitals being designed by architects. Swiss “starchitects” Herzog and de Meuron, whose work is famous for its special focus on detail, are working on the design of New North Zealand Hospital in Hillerød (Denmark). The concept of Maggie’s centres for cancer care is also well known across Britain and other parts of Europe, where famous architects (Steven Holl, Benedeta Tagliabue etc.) are designing facilities. Austria offers locally sourced design and execution, with a focus on wooden architecture and identity (Dietger Wissounig Architekten—Gesundheitseinrichtung Josefhof and others), while hospitals designed by consultants or engineering specialists that lack the architectural qualities usually connected to a humanistic approach are currently being built in Slovakia (LT Projekt, Dutch health architects, and others). Today, modern hospitals are designed with a focus on the humanization of their environments, and the needs of patients and staff are prioritized. They offer services, such as a post office, bank, hair salon, supermarket, cinema, library, and restaurants [1]. Entrance, communication, and waiting areas are designed for a quality and homey feeling. It is very economically demanding to change the principal concepts of existing hospitals; usually, it is easier to build new ones. The environmental aestheticization of existing spaces in hospitals with a focus on humanization led by skilled architects and interior designers is becoming obligatory. Moreover, there are many stereotypes connected with the methods of managing and furnishing health care facilities, especially the application of natural materials, particularly wood. 

Biophilia, defined as the affection for natural materials and nature-based concepts, works on the on the visceral, cultural, and social background of humans [2,3]. Our nervous systems know the materials, concepts, and principles originating in nature well; thus, these are viscerally related and compatible with human beings somehow. During interaction with an environmental setting built in this way, the nervous system is not irritated, and the scanning of the whole surrounding is not needed all the time. It evokes an association with survival, and the entire human organism can be in a relaxing state. On the social and cultural level, this inclination can be interpreted through the shared archetypes, since application and interaction with natural solid materials are an inseparable counterpart of all folk or traditional art and culture based on available local materials. In contemporary design and architecture, there are many material innovations, and the natural local materials are timeless and contemporary. Embracing them in their most simple form into any kind of built environment contributes to wellbeing.

Wood with its natural and emotional impact through visual, tactile, and olfactory interaction also has positive objective properties of the healthy microclimate, such as great contact comfort, improvement of room acoustics, the regulation of air humidity in a space, reduction of VOC (volatile organic compound) emissions (“sink effect”), and antimicrobial properties due to the absence of surface treatment with all kinds of surface finishing, e.g., oil, wax, or varnish. Unfortunately, without them, wood is sensible in a more extensive way to surface defect, irregular patina due to wear and obsolesce, and maintenance in a wet way is crucial. The standard use of cleaning disinfecting agents guarantees that surfaces are visually clean and without microorganisms. In the case of a “naked wooden surface”, it is not simple, but it is possible to find alternative solutions. However, is this really the case? In our culture, the belief that surface treatment with chemical agents, creating a compact film coating, can protect wood from surface defects and provide easy management and care still prevails. Trustful information should definitely be provided for users about wood surfaces’ behavior when it is in contact with any kind of fluid. On a “naked wood surface”, where wet stuff and drops are not immediately removed and wiped, water and colored fluids do leave marks because of the presence of pigments in the wood. Individual marks originating from usage and contact with colorful fluids and fats are initially not appreciated, but later, the created patina can appear as a benefit, especially if it is distributed over entire surface in “random harmony”. Application of natural oils and waxes on wood surfaces represent a compromising solution. Manufactured from renewable raw materials, surfaces processed with them have high contact comfort and are pleasant to touch. However, they significantly diminish the natural porosity and surface roughness of wood and so many of the positive qualities for a healthy indoor microclimate are reduced. Waxed surfaces have a compact coating with hydrophobic behavior. Oil surface finishings, mostly based on flax oil, can contribute naturally to the benefit of a better indoor microclimate to a certain extent, thus they surround the wood fibers. In the case of their application in more layers, as recommended for wooden work desks, flooring, and tabletops, a kind of compact, but a bit elastic, film is created. Usually, they provide complete protection against water spots and colored food and liquid stains. One of the benefits of oils and wax surface finishing is their simple method that can be done at home without professional skills and equipment; there is no necessity to remove the obsolete coating and to apply new ones in more layers, as with synthetic lacquers. If the “naked solid wood surface” starts to have an unacceptable patina, it can by renovated in a mechanical way (by usual techniques of sanding etc.), if it is a thick solid wood board or veneer with a sufficient thickness. These kinds of surfaces have not only a positive humanizing effect on indoor microclimates but can also have positive environmental impacts. The fewer additives and other chemicals are used to finalize the wooden product, the better the indoor air quality, and a positive environmental impact can be achieved.

To avoid the need for heavy additional surface treatment of solid wood, it is appropriate to select wood species according to the natural and specific durability. An appropriate zoning and technical structural protection can help too. Zoning means using wood without chemical treatment by at least those elements that are not in direct contact with fluids. Nevertheless, a very important issue is a change in the approach of professionals and users, a re-evaluation of prejudices and stereotypes, and a review of the necessity of chemical treatment of natural materials’ surfaces. This problem was broadly investigated in the research project Interaction of Man and Wood in 2013. Many collaborations with other researchers and institutions were done, including the development of super-hydrophobic surface finishes [4], the investigation of wood’s natural interaction with microorganisms according to the different surface finishings [5,6], and a comprehensive study of the suitability of wood in health care facilities. These were finally applied as solutions for hospital interiors and later into an in situ experiment.

All investigated impacts of wood to human well-being predestine wood for its use in the day, social, and health care premises and for wellness facilities—places to relax, recover, and work too. The application of wood in its natural solid form can contribute to an earlier recovery of patients in health care facilities or to general public health, not only in residential but also public spaces predestined for long-term stay.

To re-evaluate the prejudices and recommendations and hygienic standards associated with the cleaning and maintenance of surfaces, more case studies showing the behavior of natural materials in hygienic highly exposed facilities are needed. Therefore, since 2015, we have been looking for cooperation projects for the revitalization of healthcare facilities where we could implement an evidence-based design approach and design thinking. This can demonstrate that natural materials in their authentic forms, particularly wood as the most important structural indoor material, belong here, e.g., to areas, such as foyers, wait areas, and chill-out zones.

## 2. Materials and Methods 

Direct interaction and relations between human beings and a built environment is a very complex problem and needs to be explored by multi- and interdisciplinary methodology. That is the reason why many different approaches were used. The old waiting room (Figure 1a) was reconstructed to improve a patient’s well-being. Wall paneling and ceiling cladding made of solid pine wood was installed, along with a seating area made of larch timber. Secondary veneering with pinewood veneer was also used for the first time, and new warm white lighting was installed (Figure 1). The research continued in the new interior of the waiting area at the National Oncology Institute.

The microbial indoor air quality and surface contamination assessment, and analysis of VOC-emissions were executed in this space; experimental tests of the physiological responses on 50 volunteers, moving in the waiting room for 20 min, were also carried out. For the evaluation, different methods to create a complex methodology were chosen following pilot methods dealing with direct responses of human beings in an environmental setting. The EEG (electroencephalograph) (4 persons; 3 men, 1 woman, at the age of 26.3 ± 4.8; all right-handed) and the facial expressions of all our volunteers before entering, during, and after their stays in the wooden waiting room (Figure 1b) were recorded. Specifically, the ECG (electrocardiograph) (40 persons; 24 men, 16 women; at the age of 27.6 ± 9.5; 38 right-handed), heart rate (HR), and respiration activity were recorded using our self-designed ECG holter. The blood pressure and cortisol levels were observed too. The selection of the methods was based on their ability to verify the human physiological state. Brain waves change when human hormones are secreted [7,8,9]. Heart and respiration are also adaptable, and their parameters react to various environments. Sympathetic and parasympathetic activity shows the activity or tranquility of the human body [10,11,12]. Additionally, facial expressions and emotions were analyzed utilizing 4 IP (internet protocol) cameras strategically placed to cover all angles within the wooden waiting room and also the faces of the volunteers. All the devices used for this experiment were wireless, so they did not physically limit our volunteers. Simultaneously, questionnaires were used: One before entering the waiting room, with the goal of exploring the personal background of the users and their subjective reactions to natural materials in built environments; and a second one after leaving the room to ascertain feedback of the design, the wood material, and the complex comfort felt during the stay in the room (data not shown).

### 2.1. Survival of Bacteria on and in Wooden Boards and Monitoring of the Microbial Quality of the Area

To allow such a study to be conducted, the necessary laboratory pilot microbiological tests on wooden materials covered the period from January 2014 to September 2015. We chose oak and pine wood—each in three different surface treatments—the first in a naked wood surface without any kind of finishing, the second an acrylic lacquer, and the third with a hard wax oil finish (the laminated particle board was used for comparison). The top of the wooden and laminated particle board blocks (size 3 × 3 cm) was inoculated with *Staphylococcus aureus* (CCM 3953) or *Salmonella enteritidis* (CCM 4420) overnight cultures and kept in closed Petri dishes at room temperature, humidity of 40%, and a dark–light cycle. At different times (immediately, next day, week, and month, respectively), the surfaces of one sample group of the blocks were pressed against the agar plates (for 1 min), the surfaces of another sample group of blocks were swab wiped, and planed off (depth 1 mm) subsequently. Bacterial cells from the swabs and shavings were extracted by saline solution during vigorous stirring (3 × 45 s); serial dilutions were plated onto agar plates. The cultivation conditions were: Mueller–Hinton agar, 24–48 h, and 37 °C. The numbers of bacteria surviving on the surface of the blocks were evaluated as growth intensity (modified contact plates method) and/or the numbers of colony forming units (CFU; swabbing methods); the numbers of bacteria surviving in the wooden blocks were evaluated as the number of CFU (using shavings).

Passive air sampling in the area was accomplished according to the 1/1/1 scheme (for 1 h, 1 m from floor, at least 1 m away from walls or any obstacle) using Petri dishes (9 cm diameter) containing nutrient medium Tryptic Soy agar or Sabouraud agar. After cultivation at 25 °C for 2 to 5 days, the number of CFU was counted (settle plates method). Analogous culture conditions were used to determinate the presence of the viable number of microorganisms on the furniture, walls, and floor in the waiting area by the contact plates method (using 9 cm diameter Petri dishes) and swabbing method (surface area of 10 × 10 cm). The results are expressed as CFU [5,6].

### 2.2. VOC Emissions in Space

As an added value of this study, VOC emissions in the same waiting area were measured and evaluated, according to regular methods [13]. These were executed prior to the wood elements’ implementation and after 3 weeks and 4 months after the application of wooden parts of the interior, in cooperation with “Kompetenz-zentrum WOOD K plus Vienna”. The aim was to find out whether “the naked wood elements” can absorb other VOC emissions and to release them in smaller volumes into the surroundings—a so-called “sink-effect”. Two VOC samples were taken in the foyer prior to and following the reconstruction work, respectively. The samples were taken on sorption tubes filled with Tenax^®^TA for 60 min, using air sampling pumps set to a 100 mL/min air flow rate. The sorption tubes were spiked with 1 μL of internal standard toluene-d8 dissolved in methanol prior to the sampling.

The sorption tubes were subsequently thermally desorbed and evaluated by GC-MS (gas chromatography-mass spectrometry). The identification of the substances was done by comparison with commercially available spectral libraries and retention indices. The substances were quantified as toluene equivalents (TEs). All detected substances were summarized in the TVOC (total volatile organic compounds) cumulative value. The average was calculated from the duplicate samples. 

### 2.3. EEG Recording

For the measurement of brain activity, 14-channel, 14-bit EEG helmet Emotiv Epoc with saline wet electrodes and a sampling rate of 128 Hz was used (Figure 2 and Figure 3a, Table 1). Alpha waves (EEG (α) (8–12 Hz)), which are responsible for planning, motivation, solving problems, and organization [8,9], were calculated. They are recorded mostly in the frontal cortex and in the temporal lobe. Beta waves (EEG (β) (12–30 Hz)) were measured in the occipital lobe and in the parietal lobe [8]. In the central part of the brain, SMR (sensorimotor rhythm) wave EEGs (SMR) (12–15 Hz) were measured. They interpret so-called “relaxed attention”, when people are concentrated and are not in stress, so they can concentrate on long and effective work.

### 2.4. ECG Holter

The designed holter uses latest advances available in microelectronics today (Figure 3b, Table 1). The device is primarily constructed for 24/7 monitoring of ECG, therefore it can be used in various clinical and homecare applications. The core of the holter is ATxmega128A3 and Texas Instruments ADS1292R. The ATxmega128A3 is a 16-bit μ-controller with low power consumption and high performance (32 MIPS throughput at 32 MHz), which has a 128 KB flash program memory, 8 KB SRAM, and 2048-Byte EEPROM. The ADS1292R is a 24-bit ∆–Σ analog-to-digital converter, which offers 2 channels. In our case, the first channel is used for the recording of ECG and the second for respiration. The ADS1292R incorporates all features required for low-power portable devices used in sport and medical electrocardiogram applications. Power consumption of one channel is 335 μW, which is ideal for portable electronics and other battery-powered applications. The ECG holter has built-in gyroscope STMicroelectronics L3GD20, an accelerometer with magnetometer STMicroelectronics LSM303D, and a barometer with a Bosh BMP180 temperature sensor. The overall power consumption of the device is 4 mA, so the system offers an operating time over 40 h. Data are stored on the built-in SD card in CSV format. The device is connected to the body using pre-gelled disposable Ag/AgCl electrodes. In this experiment, the ECG holter was used to measure heart and respiration signals at a sampling rate of 500 Hz and acceleration at 100 SPS (samples per second).

Heart rate (HR) and HRV (heart rate variability) analysis were obtained from the ECG record [10]. The accelerometer was used only as a supplement to simplify evaluation and artefact suppression. The HR reflects the overall activity of the autonomic nervous system and this activity change provides a suitable indicator of the human state and mood. HRV is often used to recognize good health. If the HRV is higher, heart beats are more variable, and the person is more ready for action. Lower HRV is linked to increased arousal or illness. The usage of HRV analysis is widespread, not only for medical use but also for measuring emotions, thoughts, behavior, or feelings. From many studies, it is known that HRV is associated with a lot of factors. For example, an HRV increase is related to higher self-control abilities, better stress-coping abilities, and greater social skills.

From HRV analysis, the ratio of LF (low frequency) and HF (high frequency) (LF/HF) were calculated, which can demonstrate the amount of sympathetic/parasympathetic nervous system activity. Data related to the amount of LF (0.04–0.15 Hz) represents a measurement of sympathetic nervous system activity, which is activated in stress situations, during strain, escape, or attack. High-frequency HRV (0.15–0.4 Hz) correlates to parasympathetic activity, which can be an expression of regeneration processes. It provides the conditions for relaxation and rest. Low values of the LF/HF ratio mean relaxation and energy saving; vice versa, high values represent higher performance and body effort [11].

### 2.5. Face Expression

Based on Charles Darwin, facial expressions of emotions are universal for different cultures and are biologically determined (Figure 4a). Emotion expressions are not learned but the product of man’s evolution [14]. FaceMedia technologies use a facial action coding system (FACS) developed by C. H. Hjortsjö undertaken by P. Ekman and W.V. Friesen, which was first published in 1978 (with a significantly updated version published in 2002). Via FACS, humans and algorithms can manually as well as automatically code practically all anatomically possible expressions on the face. The expressions and number of action descriptors are defined with the specific action units (AU), which interpret contractions and relaxations of the muscles. Therefore, FACS can be used to specify human emotions. This tool may be used for the recognition of basic emotions or pre-programmed commands for an ambient intelligent environment [15]. Based on the 30 most important facial features and their positions, 7 types of emotion have been identified as predominant [16] (Table 2).

Universality of emotions around the world—Paul Ekman and his team conducted a series of researches and experiments in order to determine if the emotions specified by FACS are universally defined across cultures (Figure 4b). Based on a study performed in the Southeast Highlands of New Guinea with a community isolated from typical access to mass media and the outside world, the universality of the most significant displayed and identified emotional expressions was confirmed. Even communities isolated from civilization and the influences of mass media are clearly identified as FACS emotional types [17].

To transform the emotion categorization principles into detection metrics to determine the Automated Customer Satisfaction Score (ACSS^TM^) FaceMedia ACSS technology deploys a set of algorithms for continuous and real-time image processing in order to identify faces in video streams from IP cameras. To analyze the details of faces as well as facial expressions, a real-time landmark detector hat was optimized to identify 68 facial features was used. The landmark detector processes each frame separately, i.e., every frame of the video stream is processed by an algorithm in order to determine all facial landmarks for further facial feature processing.

All facial features detected and analyzed by landmark algorithms were utilized in order to identify emotion types based on the FACS. The duration of the emotions and facial expressions as well as combinations of detected emotions and facial expressions together with the accuracy of detection and probability score serves as the basis for ACSS scoring. The strategy and methodology used to measure facial expressions in the waiting area at the National Oncology Institute space is displayed in Figure 5.

### 2.6. Cortisol Analysis

Analysis of salivary cortisol was performed in all saliva samples collected from the volunteers. Cortisol was measured in unstimulated saliva using a salivary ELISA cortisol kit (DRG instruments GmbH, Marburg, Germany). The whole assay procedure was completed following the producer’s instructions. Concisely, 100 µL of non-diluted samples, standards, and low/high positive control were placed into an ELISA plate. Thereafter, 200 µL of enzyme conjugate was inserted into all wells and the whole plate was incubated on a plate shaker at 300 rpm for 1 h at room temperature. After incubation, the whole content was shaken out and washed 5 times with a washing buffer. Next, 200 µL of substrate solution was inserted and the plate was again incubated at room temperature for 30 min. The addition of 100 µL of stop solution finally stopped the reaction, and the absorbance at 450 nm was evaluated (Tecan Safire II instrument (Tecan, Männedorf, Switzerland)).

### 2.7. Statistical Analysis

Our measured data from Section 2.3, Section 2.4, and Section 2.6 were first evaluated in Labchart program (ADInstruments, Sydney, Australia). Statistical analyses were done in Microsoft Excel (Redmond, WA, USA) and in OriginLab (Northampton, MA, USA). All data were processed using GraphPad Prism 7 (GraphPad Software, Inc., La Jolla, CA, USA). After testing for normality (D’Agostino–Pearson omnibus test), data were processed with the Wilcoxon matched-pair signed rank test. Outliers were removed using the Rout test. Data are presented as mean + standard deviation (SD). The statistical significance level was set to *p* = 0.05.

## 3. Results

### 3.1. Microbiological Analysis of Wood Surfaces

Results from the laboratory tests pointed to antimicrobial activity of the wood in comparison with the laminate, where live bacteria were observed (actually, they were there also after 20 h). Higher antimicrobial activity was observed in oak block compared to the pine wood block. Contrary to laminate, wooden blocks showed no live cells after 20 h on the oak block in the case of *Staphylococcus aureus* or less than 5% live bacteria on the pine block. The swabbing method was used for this analysis. It is interesting that the bacteria displayed a higher survival on the lacquered wood compared to only the oil-treated wood, where less bacteria was found [5,6].

Lab tests showed that the oak and pine without chemical finishing had higher antimicrobial effects due to the presence of tannins and terpenoids, respectively. Tests focused on the microbial quality of indoor air and surface contamination assessment analyzed in the original state before reconstruction and the current state three weeks after the installation of wooden elements in the revitalized part of the foyer at the National Oncology Institute. In comparison to the original state, after the implementation of wood elements, a much better state was found while the indoor air appeared unaffected. Therefore, a high level of hygiene standard was found (Table 3 and Table 4). Table 5 shows the overall number of colony-forming units of microorganisms on the wooden surfaces, determined by the standardized method of contact plates.

According to the state after 3 months and 6 months of regular usage for sitting, there was a significant presence of dirt spots on the solid wood surfaces without finishing. After microbial investigation of the spots, it was confirmed that they have no microbiological character, but since the layer of dirt diminishes the natural antimicrobial ability of wood, and not only for aesthetic reasons, it is necessary to find a solution for suitable cleaning. The crucial point in the application of solid wood without chemical treatment is its maintenance. An unusual way of cleaning was proposed—the application of ethanol-technical alcohol in comparison to the usual chlorine-based detergent “Savo”. The results displayed in Table 6 show that cleaning with alcohol is more efficient for diminishing the visual presence of dirt and renewing the natural antimicrobial properties of solid wood without finishing (Figure 6). The regular cleaning method using a sponge penetrated with the ethanol and SAVO and pressing it towards the surface was used. 

### 3.2. Results of VOC-s Measuring

According to the report from WOOD Kplus, BOKU Vienna, the VOC detected in the indoor air of the foyer prior to the reconstruction work belonged to different substance groups. Most of these substances are found in solvents and cleaning agents. The highest concentrated substances are octanal, 2-Ethylhexanal, limonene, nonanal, xylenes, and decanal. After the reconstruction work was finished, the VOC profile changed. Amounts of VOC were detected in the tested area. These substances belong to the group of terpenes and showed the highest concentration in the indoor air. Three weeks after the reconstruction work was finished, the main substances were α-pinene, β-pinene, limonene, and Δ-3-carene. Three months later, α-pinene and Δ-3-carene showed the highest concentration in the indoor air of the foyer. However, their concentration declined compared to the results obtained three weeks after the reconstruction. Table 7 shows the overall VOC emissions evaluated in the waiting area prior to the reconstruction and 3 weeks and 3 months after.

The substances nonanal, decanal, and 2-ethylhexanol, which showed an elevated concentration prior to reconstruction work, were reduced by 62%, 78%, and 92%, respectively, three weeks after the reconstruction work was finished. Octanal could no longer be detected. Xylene concentrations remained at a comparable level, whereas limonene clearly increased after the installation of wooden furnishings. Three months later, nonanal, decanal, and 2-ethylhexanol were still detected in low concentrations in the renovated area. However, parallel samples taken in a non-renovated area of the foyer showed comparably low values for these substances. Limonene, which is not only emitted by wood but also widely used as a fragrance in cleaning agents, was exempted. The limonene concentration was 43% lower in the renovated area than in the non-renovated area. To obtain research data about the sink effect with high research relevance, it is necessary to simulate a controllable situation with indoor air conditions in a laboratory environment. VOC emissions, such as xylen, octanol, 2-ethylhexanol, etc., are compounds that form part of cleaning and disinfection agents. Artificial materials used for built environments that are regularly used in health care environments and in higher concentrations that are harmful to human health include: α-Pinene, D-3-carene, and limonene, which are terpenoids—typical of coniferous wood species. To explore this phenomenon, further measurements are necessary. The reconstruction work and the installation of wood in the foyer clearly led to a change in the VOC profile. In addition to the substances already measured prior to the reconstruction work, wood-specific VOC were identified. This also led to an increase of the total VOC concentration. However, prior to the reconstruction work, the TVOC (total volatile organic compound) was rather low.

### 3.3. EEG Recording 

EEG analysis showed a decline in EEG (α) and (EEG (β) waves in the new wooden waiting room. EEG (α) decreased by −17% (*p* = 0.25), EEG (β) by −13% (*p* = 0.44), and SMR waves by −7% (*p* = 0.88). The time progress (comparison of first 10 min versus the second 10 min) in the wooden waiting room showed a decrease in alpha waves by −4% (*p* = 0.25), an increase in beta waves by +36% (*p* = 0.13), and a decrease in SMR waves by −65% (*p* = 0.13) (Figure 7). A comparison of the hemispheres showed more significant differences in the case of SMR waves. The SMR waves were more affected on the left side (decrease by 13%, *p* = 0.13) in comparison with the right hemisphere (decrease by 2%, *p* = 0.22) (Figure 8). The wood affects the sensomotoric cortex in the left hemisphere only, which controls analytic thinking and logic. The right hemisphere, controlling emotions, art, and creativity, showed uniform activity during waiting room stays.

### 3.4. ECG Measurement 

ECG data showed a decrease of the LF/HF ration by 26.4% (*p* = 0.06) in the wooden waiting room. HR and RF (respiration frequency) in the old waiting room were 77.56 bpm and 18.14 bpm, respectively, and in the wooden waiting room it was 78.26 bpm and 18.67 bpm (Figure 9), respectively. Modest increase of the HR by 0.9% (*p* = 0.59) and RF by 2.8% (*p* = 0.09) in the wooden waiting room in comparison with the old waiting room was monitored. The time progression in the wooden waiting room showed no significant changes of HR, RF, or LF/HF.

### 3.5. Blood Pressure 

Blood pressure saw a modest decrease from average levels of 127.3/83.54 ± 14.9/14 to average levels of 126.8/82.6 ± 15.2/9.99, which is a decrease of 0.4% (*p* = 0.68) in systolic pressure and a decrease of 1.2% (*p* = 0.08) in diastolic pressure (Figure 10).

### 3.6. Face Emotion Detection

From the camera records, noticeable positive mood increases of +0.16 were observed in the automated customer satisfaction score, which equals a 7.65% enhancement in comparison to the automated customer satisfaction score before entering the wooden waiting room. Detailed levels of the emotions during stays in the wooden waiting room are visible in Figure 11.

Very satisfied and satisfied emotions are dominant, with a 46% share of all detected emotions, followed by neutral emotions (42%). Somewhat satisfied and not satisfied emotions detected on the basis of the facial expressions of volunteers represent only a 12% share.

Detailed progress of the emotions (Figure 12) clearly indicates and correlates with the ACSS scoring methodology. The first section of the graph indicates a progression of emotions during the entry to the environment phase. There is a significant increase in the disgust and contempt emotions. Once the participants entered (second section of the graph), surprise, happy, and neutral emotions were observed as the main driver of the positive attitude towards the wooden waiting room environment. The last section (exit) highlights positive expressions and feelings from the wooden waiting room interior, with a predominance of happy and surprise emotions.

### 3.7. Cortisol 

No significant differences were found between cortisol in before and after samples. However, the *p* value was *p* = 0.13, therefore, it is a moderate tendency towards a decreasing concentration of cortisol as a result of the influence of the environment. The average level of the cortisol before entering the waiting room was 2.43 ± 1.76 ng/mL. After a stay in the waiting room, the average cortisol level was 2.26 ± 1.61 ng/mL. Accordingly, the cortisol level decline was 7.5% (Figure 13. Therefore, it can be suggested that wood has an anti-stress impact on humans.

## 4. Discussion

Environmental settings built with a prevalence of natural materials, particularly wood, affects the user´s nervous system in a positive way. It contributes to stress processing and can contribute to shortening the patients’ treatment, as demonstrated by several global studies [21,22,23,24,25,26]. Moreover, our results and findings support these findings.

Following and re-evaluating different studies of the interaction of wood and microorganisms [27,28,29], in 2014, our laboratory microbiological research started with *Salmonella enteritidis* and *Staphylococcus aureus*. These lab tests showed that oak and pine wood without chemical finishing showed immediately better antimicrobial effects [6], due to the presence of tannins in the oak and terpenoids in the pine. The phenomenon is not present in all tested surface chemical treatments. Natural oil finishing has a slightly better performance than acrylic paint, while laminated particle board had the lowest antimicrobial effect and the optimal conditions for survival of bacteria over several days. These tests showed that wood without surface treatment has the strongest antimicrobial effects. There is a hypothesis that due to the anatomical structure of wood, all kinds of wood do not support the reproduction of all kinds of micro-organisms. However, our microbial test in the waiting area in situ showed that after 4 and 7 months of intensive usage in a foyer, a public space without intensive cleaning care, the layer of dirt reduced the natural antimicrobial effect. Those parts that were not in intensive contact with users (e.g., wood paneling on the walls) maintained their initial antimicrobial effect. After 4 and 7 months, the microbial quality of the air was better in comparison to the state prior to reconstruction. Thanks to chemical cleaning with ethanol and regular mechanical cleaning with brushes or sand machines, it is possible to maintain acceptable hygiene standards.

One other option is to zone the inner spaces of health care facilities to use wood and other naturally porous natural materials only in less exposed areas where they can contribute in a positive way to the microclimate and well-being. To achieve an acceptable level of standard cleaning, it is possible to use oils, waxes, and lacquers on intensively used surfaces, while leaving other surfaces without finishing. In the measurements of VOC emissions, a hypothesis that wood has a so-called “sink effect” [30] has been presented, where certain building materials used indoors, including solid wood, have the ability to absorb VOC emissions and are desorbed in a smaller volume. With respect to VOC emissions in our tests, a decrease in a number of substances connected with cleaning agents or solvents could be observed in the renovated area of the foyer three weeks after the finishing of the reconstruction. This could possibly be an effect of the installation of wooden furnishings. Wood has the potential for the adsorption and temporally delayed re-emission of VOC from other sources. However, three months after the reconstruction work was finished, the concentrations of the relevant substances in the non-renovated area were comparable to those in the renovated area. To clearly prove the adsorptive effect of the wooden furnishings, further measurements in areas with lower ventilation rates would be necessary.

Apart from the positive antimicrobial effect and better air in the waiting room, wood materials have a considerable impact on human physiology. The wireless devices for the higher comfort of our volunteers were chosen [31,32]. This article wants to show the importance of an environment on human well-being. The recording of human brain waves showed total decreases of brain activity in a wooden waiting room in comparison with the former space. EEG (α) decreased by −17%, EEG (β) by −13%, and SMR waves by −7%. The effect of wood on the nervous system showed that the brain becomes calmer and less stressed, probably because wood is natural and more familiar for humans. The first effect of the wood during the first few minutes was relaxing, while during longer stays in the wood environment, the brains of our volunteers became more active, less afraid and nervous, and their memory and ability to think increased. It is justified by a decrease in alpha waves by −4%, an increase in beta waves by +36%, and a decrease in SMR waves by −65%. It is because wood is more interesting for human perception. Consequently, the EEG (β) waves are higher during a longer stay in a wood environment [8] and EEG (α) waves are also lower when people have visual stimuli [33]. Wood smells good and has various patterns [34]. Wood affects EEG (SMR) waves differently in the hemispheres of the brain. In the left hemisphere, which controls analytic thinking and logic, these waves were significantly lower than in the right hemisphere. The right hemisphere controls emotions, art, and creativity, and was not decreased to such an extent. Therefore, wooden materials suppress thinking activities and let creativity work.

Miyazaki et al. [34] prepared experiments in various forest environments and discovered decreases in cortisol levels, sympathetic nervous activity, and systolic and diastolic blood pressure. A mild increase in HR and RR was registered in the environment with wood installed. This could be caused by shorter stays in the wood environment and the “wow effect” of the first contact with wood. Longer stays may have a decreasing effect on the HR. Parasympathetic nervous activity was enhanced, indicating a relaxed state and reductions of energy expenditure. In our experiment, a similar physiological activity in comparison between wood and no wood environments was observed. Cortisol levels also declined, therefore, showing a tenuous anti-stress effect. In addition, sympathetic activity decreased and parasympathetic was enhanced, which means that our volunteers felt more relaxed in the wood environment [11,35].

## 5. Conclusions

Initial experiments in an actual hospital environment—concretely in a waiting room at the National Oncology Institute—were designed to maximize the benefits of using natural materials. Wood has a good influence on patients, and its antimicrobial ability maximizes the benefits of this design. According to our results, the hypothesis that “the naked wood surface” in direct regular contact with man gets dirty and the antimicrobial effect reduction in this state was confirmed. However, surfaces in rare direct contact had the same antimicrobial effect and the same positive effect on indoor air quality. The goal of this study was also to optimize and recommend the cleaning and care process and for the zoning of “the naked wood surface” application. 

The presented results confirmed the relation of the material and human physiology. The physiological parameters of our volunteers showed positive reactions of the bodies from the environment with wood materials and a “wow effect” of the wood on humans. The brain is under less stress, the natural environment is probably more familiar for the brain, or it has fewer stimuli. Wood influences the sensomotoric cortex mostly on the left hemisphere, which is responsible for analytic thinking and logic. Even though there was a hypothesis that the right hemisphere will be more activated due to being relaxed, this can be confirmed just after a longer time spent in the space. From the results of the applied different pilot methods, it is possible to conclude that all of them showed responses and feedback from respondents to the different environmental settings, while a different exposure time was needed for the synchronization to reach a critical point. In this way, the methods can be improved and more appropriate for our future studies. There are not many articles about methods that are used to the measurement of the environment’s effect on humans, but they could be really relevant tools for identifying human well-being. 

Another option is to zone the interior spaces of health care facilities and to use wood and other naturally porous materials only in less exposed areas where they can contribute in a positive way to the microclimate and well-being. To maintain the level of standard cleaning, it is possible to use oils, waxes, and lacquers on the intensively used surfaces but to leave other surfaces without finishing. The wood can be recommended for hygienically less exposed areas of health, social, and day care premises, such as ceiling and wall paneling in any kind of indoor spaces that do not come into direct contact with colored and contaminated fluids. This could be a way to change the prejudices and stereotypes common in building and furnishing health, social, and day care facilities.

It is necessary to educate architects and interior designers about the benefits of using wood not only in hospital spaces but also in education and working facilities. These are spaces where people are more exposed to stress. Interiors designed on principles of humanization can help with reducing such negative effects. For example, the length of patients’ time spent in hospital can be reduced by using the properties of wood. On the other hand, a tailored design can be an expensive solution, so the development of wooden products and interiors should be more oriented to standardization. The results of this research could help with propagation and marketing for introducing more natural wood in hospitals.

This paper has shown experiences with a complex methodology, and presented results showing how the application of solid wood contributes to well-being and is suitable for health care and social and day care facilities along with appropriate zoning and cleaning.

## Figures and Tables

**Figure 1 ijerph-16-03496-f001:**
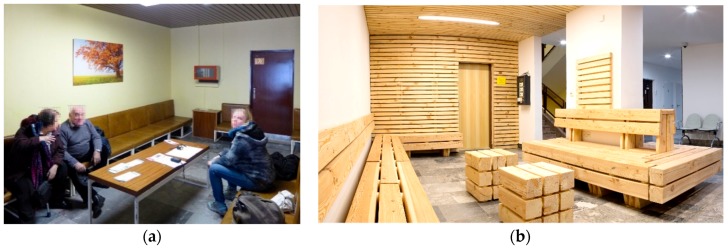
Wooden waiting room at National Oncology Institute: (**a**) prior to reconstruction; (**b**) after reconstruction in 2016.

**Figure 2 ijerph-16-03496-f002:**
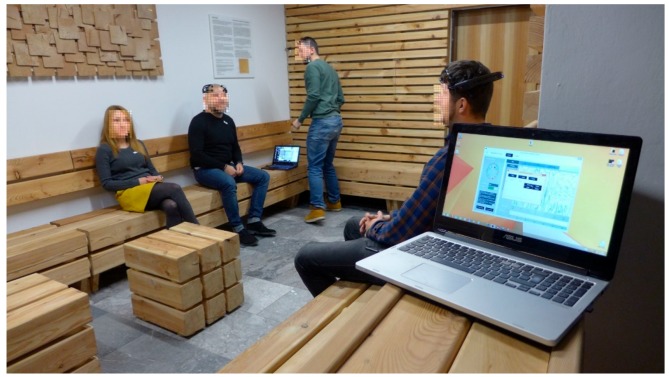
Experiment—usage of Epoc plus and ECG (electrocardiography) holters.

**Figure 3 ijerph-16-03496-f003:**
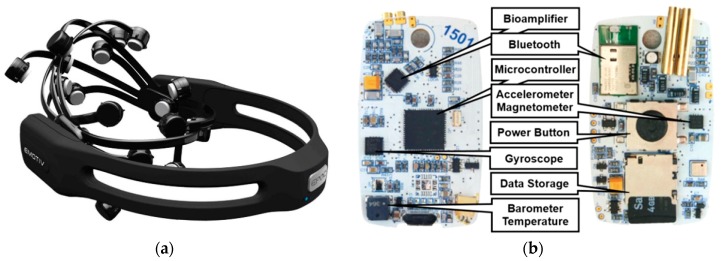
Emotiv Epoc helmet (**a**) and designed ECG holter (**b**).

**Figure 4 ijerph-16-03496-f004:**
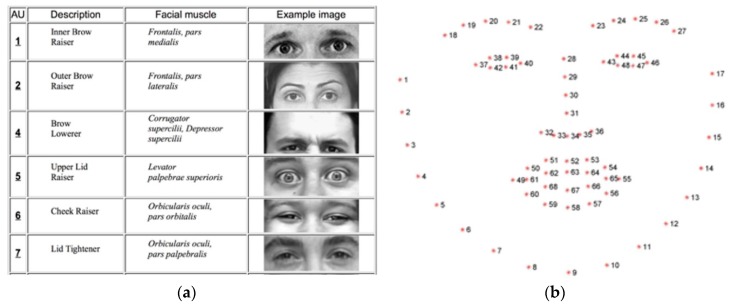
Examples of identified facial features (**a**) and facial features detected by the FaceMedia landmark detector (**b**).

**Figure 5 ijerph-16-03496-f005:**
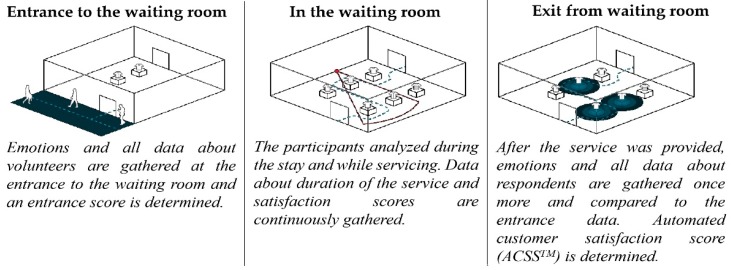
Face expression methodology.

**Figure 6 ijerph-16-03496-f006:**
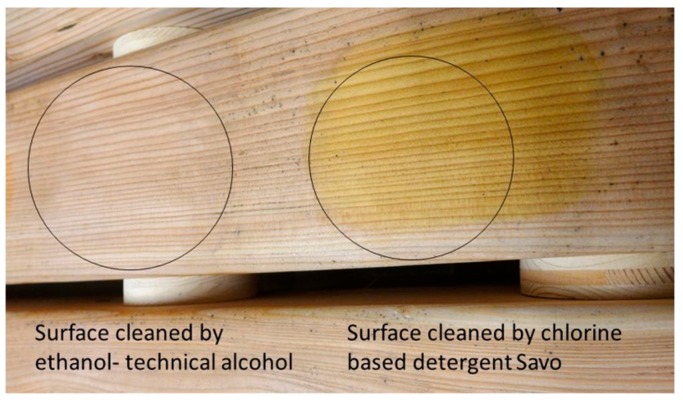
Results of different approaches to cleaning solid wooden surfaces; the left circle is cleaning with ethanol and the right circle is cleaning with chlorine detergent.

**Figure 7 ijerph-16-03496-f007:**
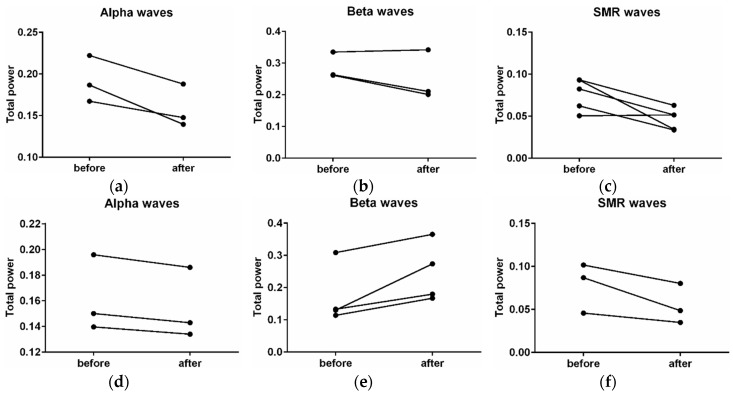
Changes in the activity of alpha, beta, and SMR (sensorimotor rhythm) waves before stays in the wooden waiting room and after stays in the wooden environment (**a**–**c**). Changes in the activity of alpha, beta, and SMR waves in the wooden waiting room at the first third of the time spent in the wood environment (before) in comparison with the third third of that time (after) (**d**–**f**).

**Figure 8 ijerph-16-03496-f008:**
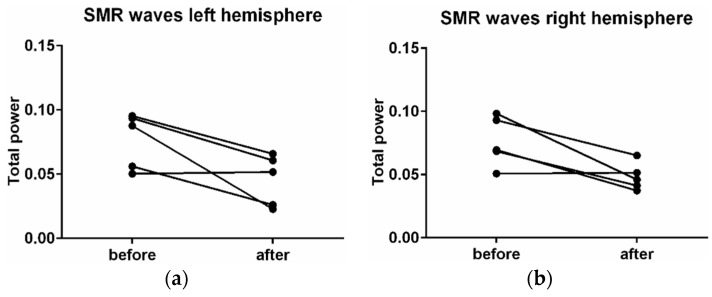
Changes in activity of the SMR waves in the left (**a**) and right (**b**) hemisphere.

**Figure 9 ijerph-16-03496-f009:**
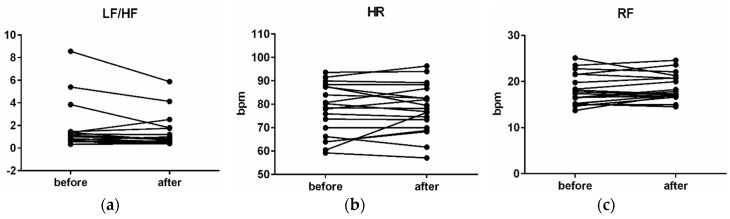
Changes in the LF/HF (low frequency/high frequency) (**a**), HR (heart rate) (**b**), RF (respiration frequency) (**c**), before stays in the wooden waiting room and after stays in the wood environment.

**Figure 10 ijerph-16-03496-f010:**
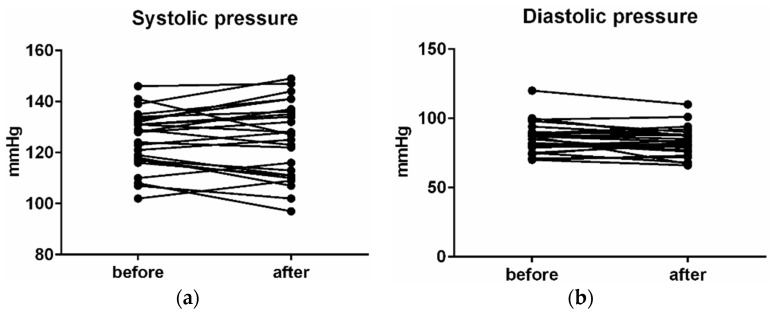
Changes in activity of the SMR waves on left (**a**) and right (**b**) hemisphere.

**Figure 11 ijerph-16-03496-f011:**
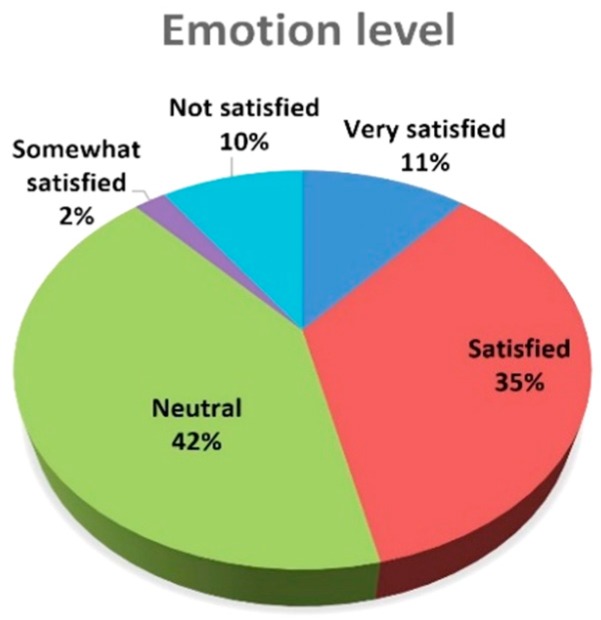
Level of the emotions in the wood environment of the waiting room (during stay).

**Figure 12 ijerph-16-03496-f012:**
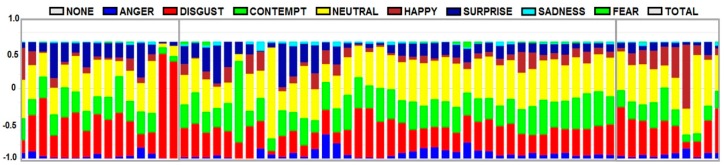
Progression of emotions at the entrance, during the stay, and the exit from the waiting room.

**Figure 13 ijerph-16-03496-f013:**
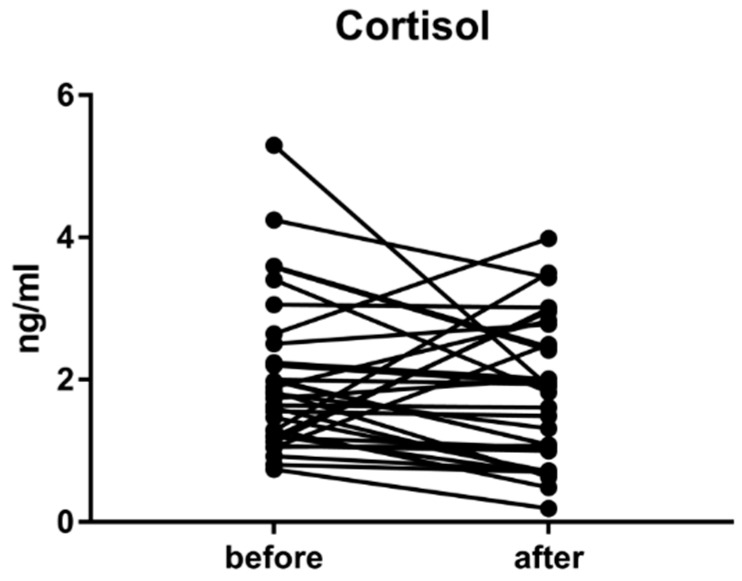
Changes in cortisol level before a stay in the wooden waiting room and after staying in a wood environment.

**Table 1 ijerph-16-03496-t001:** Technical parameters of the EEG helmet and ECG holter.

EEG Helmet	
No. of Channels	14 AF3, AF4, F7, F3, FC5, FC6, F4, F8, T7, T8, P7, P8, O1, O2
References	CMS/DRL located in P3 and P4
Sampling method	Sequential sampling. Single ADC
Sampling rate	128 SPS
Resolution	14 bits
Filtering	Band pass filter 0.2–43 Hz, 50/60 Hz notch
	Order: 5th
Dynamic range	8.4 mV (pp)
Coupling mode	AC coupled
Built-In	2-axis gyroscope
Connectivity	2.4 GHz Bluetooth^®^ Smart
Output format	EDF
Battery	Li-Pol 640 mAh
Battery life	up to 6 h using
Electrodes	Saline-based wet sensors
**ECG Holter**	
Bioamplifier	TI ADS 1292R
Number of Channels	2 Low-Noise PGAs
Resolution	24-bit, no data missing
Sampling rates	125–8000 SPS
Built-In	Respiration Measurement
Microcontroller	ATxmega 128A3
Performance/Speed	16-bit AVR, 32 MHz
Acceleration sensor	STMicroelectronics LSM303D
Ranges	±2–±16 g (3D)
Resolution/Sampling rate	16 bit (3–1600 SPS)
Magnetometer	STMicroelectronics LSM303D
Ranges	2–12 G (3D)
Resolution/Sampling rate	16-bit (3–100 SPS)
Gyroscope	STMicroelectronics L3GD20
Ranges	250–2000 dps (3D)
Resolution/Sampling rate	16-bit (95–750 SPS)
Barometer	Bosh BMP180
Ranges	300–1100 hPa
Resolution/Sampling rate	16-bit (2–50 SPS)
Temperature	Bosh BMP180
Range/Accuracy	0–65 °C/±1 °C
Output data format	CSV, EDF+
Supply voltage	Li-Pol 120 mAh
Battery life	up to 40 h (on 250 SPS)
Connectivity	Micro USB
Optional	2.4 GHz Bluetooth^®^
Electrodes	Disposable Ag/AgCl
Dimensions	37 × 25 × 15 mm
Next features	LED and acoustic signalization, trigger button

**Table 2 ijerph-16-03496-t002:** Types of detected emotions.

**Happiness**—is the great emotion that most people want to feel. Happy people feel good. Commonly, people feel this emotion when they experience enjoyable sensations. An important part of happiness is excitement, the opposite of boredom, which is what people feel when they discover something new that arouses their interest.
**Surprise**—is a brief emotion that appears when something unexpected happens. Additionally, it disappears in a moment. If a person anticipates an event, he/she cannot be surprised. Also, if people presume the surprising event, they are not surprised.
**Fear**—this is a more negative emotion, in which people fear harm. That may be physical or/and psychological. Psychological harm can also vary from minor insults or disappointment to extreme assaults.
**Anger**—is the most dangerous emotion. Anger can arise from frustration resulting from interference, or rage from subjective feelings of the individual or the objective behavior of other people or from similar emotion states.
**Disgust**—is an aversion emotion in which people feel as if they want to spit out something, or they feel as if they have eaten something distasteful, or seen something related to an abhorrent event, etc.
**Sadness**—is internalized suffering. Most often, it is the emotion of loss—the loss of an opportunity or the reward that should follow, negative circumstances, or another’s disregard. The face is pale; the muscles flaccid, the eyelids drooped.
**Contempt**—it is a mixture of disgust and anger (not classified by Ekman but often used as an important component of emotion analysis methodologies). People experiencing contempt judge that they are more than others.

**Table 3 ijerph-16-03496-t003:** Overall number of colony-forming units (CFU) of microorganisms in the indoor air (evaluated from microbial fallout onto Petri dishes left open to the air, determined by settle plate methods). The highest permissible concentration of dust and microbial particles in clean facility spaces is 500 CFU/m^3^ [18] (that is 175 CFU/dm^2^/h, <10 CFU/m^3^ – <350 CFU/m^2^/h [19]).

	State Prior to Wood Installation May 2016	State 3 Weeks after Wood Installation November 2016	State 4 Months after Wood Installation March 2017	State 7 Months after Wood Installation June 2017
Air in the corner of the lobby	51 CFU/dm^2^/h	20 CFU/dm^2^/h	36 CFU/dm^2^/h	42 CFU/dm^2^/h
Near to the column by the sitting area	56 CFU/dm^2^/h	24 CFU/dm^2^/h	-	38 CFU/dm^2^/h
Near to the corridor to the clinics	30 CFU/dm^2^/h	27 CFU/dm^2^/h	33 CFU/dm^2^/h	36 CFU/dm^2^/h

**Table 4 ijerph-16-03496-t004:** Overall number of colony-forming units (CFU) of microorganisms on the surfaces (determined by the swabbing method).

	State Prior to Wood Installation May 2016	State 3 Weeks after Wood Installation November 2016	State 4 Months after Wood Installation March 2017
Sitting elements (seat)	6 CFU/cm^2^	3 CFU/cm^2^	12 CFU/cm^2^
Sitting elements (back lean)	˃15 CFU/cm^2^	1 CFU/cm^2^	1 CFU/cm^2^
Bench (gap between single timbers)	-	-	2 CFU/cm^2^
Table with magazines	˃15 CFU/cm^2^	4 CFU/cm^2^	6 CFU/cm^2^
Stone floor	˃15 CFU/cm^2^	5 CFU/cm^2^	7 CFU/cm^2^
Wood paneling on the wall (1 m from floor)	-	2 CFU/cm^2^	3 CFU/cm^2^
Wood paneling on the wall (1.5 m from floor)	-	-	3 CFU/cm^2^

**Table 5 ijerph-16-03496-t005:** Overall number of colony-forming units (CFU) of microorganisms on surfaces (determined by contact plates method). The limit is <5 CFU/cm^2^ [20].

	State after Reconstruction November 2016	State after Reconstruction March 2017	State after Reconstruction June 2017
Chair/bench (sitting part)	2 KTJ/cm^2^	<1 KTJ/cm^2^	1 KTJ/cm^2^
Table for newspapers	1 KTJ/cm^2^	1 KTJ/cm^2^	1 KTJ/cm^2^
Wall	<1 KTJ/cm^2^	<1 KTJ/cm^2^	1 KTJ/cm^2^

**Table 6 ijerph-16-03496-t006:** Testing of the surface by the swabbing method in relation to the possibilities of cleaning and maintenance 4 months after installation of wooden elements.

Bench seat, most frequently used	25 CFU/cm^2^
Bench seat after cleaning with ethanol	2 CFU/cm^2^
Bench seat after disinfection with SAVO (disinfection agent based on sodium chloride)	21 CFU/cm^2^
Plastic chair in the nearby part of foyer	30 CFU/cm^2^
Plastic chair after disinfection with ethanol	3 CFU/cm^2^

**Table 7 ijerph-16-03496-t007:** VOC (volatile organic compound) emissions evaluated in the waiting area.

Substance	Prior to Reconstruction	3 Weeks after Reconstruction	3 Months After Reconstruction—Renovated Area	3 Months after Reconstruction—Non-Renovated Area
Xylene [μg·m^−3^TE]	4	5	3	2
Octanal [μg·m^−3^TE]	3	n.d.	n.d.	n.d.
2-Ethylhexanol [μg·m^−3^TE]	6	<1	1	1
Nonanal [μg·m^−3^TE]	11	4	5	4
Decanal [μg·m^−3^TE]	9	2	5	4
α-Pinene [μg·m^−3^TE]	n.d.	59	50	n.d.
Δ-3-Carene [μg·m^−3^TE]	n.d.	12	6	n.d.
Limonene [μg·m^−3^TE]	2	25	7	12

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
