# Peer review of "Wood and Its Impact on Humans and Environment Quality in Health Care Facilities"

_ijerph, 2019, doi:10.3390/ijerph16183496_

Round 1

Reviewer 1 Report

The paper describes how wood can be used as cladding in health care facilities, moreover its impact on humans and environment quality is analyzed.

In the introduction part, the literature review should be implemented carefully. Too many sentences without appropriate references are reported.

Line 184: how Staphylococcus Aureus was distributed on the surfaces?

The methodology was described in a correct and precise way. VOC values are subjected to the statistical analysis reported in 2.7 or just the data collected in 2.3-2.6 are analyzed using what described in 2.7? Could the authors specify as well the uncertainty of the method used to measure VOC?

Are there any reference to standard test methods applied? If yes, please cite them.

Line 340: Please can you specify better how the cleaning steps were performed?

Author Response

Dear reviewer,

we revised our manuscript according to your report:

Point 1: In the introduction part, the literature review should be implemented carefully. Too many sentences without appropriate references are reported.

Response 1: References have been added. In the introduction there is a lot of original statements and conclusions of own research.

Point 2: Line 184: how Staphylococcus Aureus was distributed on the surfaces?

Response 2: The method was added.

Point 3: The methodology was described in a correct and precise way. VOC values are subjected to the statistical analysis reported in 2.7 or just the data collected in 2.3-2.6 are analyzed using what described in 2.7? Could the authors specify as well the uncertainty of the method used to measure VOC?

Response 3: Just the data collected in 2.3, 2.4 and 2.6 are subjected to the statistical analysis reported in 2.7.

Point 4: Are there any reference to standard test methods applied? If yes, please cite them.

Response 4: The methods of measuring of VOC emissions are cited in the literature number [29].

Point 5: Line 340: Please can you specify better how the cleaning steps were performed?

Response 5: It was more specified in methods.

Reviewer 2 Report

The paper presents an interesting research about the use of wood in health or social facilities.

The manuscript presents an overview on wood use as natural material for furniture and finishing element for internal spaces; different properties of timber has been taken into account: aesthetics as colour/texture; touch, acoustic, olfactory comfort; properties of wood to regulate the air humidity; volatile organic compound emissions (VOC-emissions). The original results presented by the authors are the experimental tests of physiological responses have been conducted on volunteers that stayed in the studied room.

General observations

The paper is complete and clear but in some parts sentences and concepts are explained in a non-scientific way:

- Please avoid to use the “we” personal form. E.g. In the Abstract (the use of “we” can be found all the article long) line #25; #27; #28

- Please define all the acronyms (e.g. ECG; EEG etc…)

- Please avoid “personal opinion” such as: #48-49; #50-52; #70-71

1. Introduction

Regarding the use of untreated surface of wood, sentence #106-109 seems to be inappropriate, in fact “fluids and fat staff” represent dirty ad the same authors write few line below #120 “If the “naked solid wood surface” is getting dirty and has unacceptable patina, it can by renovated in a mechanical way

4. Discussion

#477-479 Please try to justify this important observation with the data collected during the experiments otherwise it can appear a conclusion not based on scientific observation.

#480-482 same observation

5 Conclusions

#519 -524 Please avoid this sentence or rewrite it (as it was written, it seems just an authors’ opinion)

#541-543 “this paper has shown our experience” please do not conclude in this “personal” way a scientific paper.

Author Response

Dear reviewer,

we revised our manuscript according to your report:

Moderate English changes required.

We changed some not correct English sentences.

Point 1: Please avoid to use the “we” personal form. E.g. In the Abstract (the use of “we” can be found all the article long) line #25; #27; #28

Response 1: The corresponding sentences have been reformulated.

Point 2: Please define all the acronyms (e.g. ECG; EEG etc…)

Response 2: Acronyms were defined: ECG – electrocardiograph, EEG – electroencephalograph, IP – internet protocol, GC-MS - Gas chromatography–mass spectrometry, SMR – sensorimotor rhythm, SPS – samples per second, HRV – heart rate variability, RF – respiration frequency

Point 3: Please avoid “personal opinion” such as: #48-49; #50-52; #70-71

Response 3: The corresponding sentences have been reformulated and personal opinions deleted.

Point 4: Regarding the use of untreated surface of wood, sentence #106-109 seems to be inappropriate, in fact “fluids and fat staff” represent dirty ad the same authors write few line below #120 “If the “naked solid wood surface” is getting dirty and has unacceptable patina, it can by renovated in a mechanical way

Response 4: It was reformulated to be clear in the statement.

Point 5: #477-479 Please try to justify this important observation with the data collected during the experiments otherwise it can appear a conclusion not based on scientific observation.

Response 5: A sentence with specific numbers about the change of EEG parameters was added for confirmation.

Point 6: #480-482 same observation

Response 6: Next sentence with specific numbers about the change of EEG parameters was added.

Point 7: #519 -524 Please avoid this sentence or rewrite it (as it was written, it seems just an authors’ opinion)

Response 7: These sentences were changed and shortened.

Point 8: #541-543 “this paper has shown our experience” please do not conclude in this “personal” way a scientific paper.

Response 8: These sentences were changed.

Reviewer 3 Report

My comments on the paper:
- The goal of the paper and main results are included in the abstract.
- Keywords are appropriate.
- The structure of the paper is appropriate.
- The introduction provides the necessary background information.
- The research methodology used by the author is adequate for the approached subject
- The results of the research are clearly underlined.

In my opinion, the materials and metods section should be expanded:
- Data on the group of volunteers, age and gender should be completed. This is important because different age groups experience the environment differently.
- The EEG study of four volunteers does not give reliable results.

Author Response

Dear Reviewer,

we revised our manuscript according to your report:

Point 1: Data on the group of volunteers, age and gender should be completed. This is important because different age groups experience the environment differently.

Response 1: Data of the volunteers group were added.

Point 2: The EEG study of four volunteers does not give reliable results.

Response 2: Unfortunately, physical implementation (especially EEG) was very time consuming. Only 2 EEG devices were available at the same time. Reservations in the functional hospital were limited in time. Some EEG data was incomplete, signals interrupted. Therefore, out of a total of 8 EEG records, we managed to process only 4 data reliably. We understand that this is not a sufficiently reliable group. But we would like to give these data at least as indicative as they have been measured.
